# Temperature, traveling, slums, and housing drive dengue transmission in a non-endemic metropolis

**Juan Manuel Gurevitz**[1]*, **Julián Gustavo Antman**[2], **Karina Laneri**[3], **Juan Manuel Morales**[1]

**1** Grupo de Ecología Cuantitativa, INIBIOMA-CONICET, Universidad Nacional del Comahue, S.C. de Bariloche, Río Negro, Argentina, **2** Gerencia Operativa de Epidemiología, Ministerio de Salud de la Ciudad Autónoma de Buenos Aires, Buenos Aires, Argentina, **3** Grupo de Física Estadística e Interdisciplinaria, CONICET, Centro Atómico Bariloche-CNEA, S.C. de Bariloche, Río Negro, Argentina

* jmgurevitz@comahue-conicet.gob.ar

## Abstract

Dengue is steadily increasing worldwide and expanding into higher latitudes. Current non-endemic areas are prone to become endemic soon. To improve understanding of dengue transmission in these settings, we assessed the spatiotemporal dynamics of the hitherto largest outbreak in the non-endemic metropolis of Buenos Aires, Argentina, based on detailed information on the 5,104 georeferenced cases registered during summer-autumn of 2016. The highly seasonal dengue transmission in Buenos Aires was modulated by temperature and triggered by imported cases coming from regions with ongoing outbreaks. However, local transmission was made possible and consolidated heterogeneously in the city due to housing and socioeconomic characteristics of the population, with 32.8% of autochthonous cases occurring in slums, which held only 6.4% of the city population. A hierarchical spatiotemporal model accounting for imperfect detection of cases showed that, outside slums, less-affluent neighborhoods of houses (vs. apartments) favored transmission. Global and local spatiotemporal point-pattern analyses demonstrated that most transmission occurred at or close to home. Additionally, based on these results, a point-pattern analysis was assessed for early identification of transmission foci during the outbreak while accounting for population spatial distribution. Altogether, our results reveal how social, physical, and biological processes shape dengue transmission in Buenos Aires and, likely, other non-endemic cities, and suggest multiple opportunities for control interventions.

## Author summary

Dengue fever is mainly transmitted by a mosquito species that is highly urbanized, and lays eggs and develops mostly in artificial water containers. Dengue transmission is sustained year-round in most tropical regions of the world, but in many subtropical/temperate regions it occurs only in the warmest months. To improve understanding of dengue

**Data Availability Statement:** Data on dengue cases cannot be shared publicly because of restrictions of the Ministry of Health of the City of Buenos Aires. Access to these data can be

requested at https://miba.buenosaires.gob.ar.
Demographic data is available for download from
https://redatam.indec.gob.ar/. Weather data can be
solicited to cim@smn.gov.ar.

**Funding:** This work was supported by a grant from
Agencia Nacional de Promoción Científica y
Tecnológica (PICT 2017-0787; http://www.agencia.
mincyt.gob.ar). The funders had no role in study
design, data collection and analysis, decision to
publish, or preparation of the manuscript.

**Competing interests:** The authors have declared
that no competing interests exist.

transmission in these regions, we analyzed one of the largest outbreaks in Buenos Aires
city, a subtropical metropolis. Based on information on 5,104 georeferenced cases during
summer-autumn 2016, we found that most transmission occurred in or near home, that
slums had the highest risk of transmission, and that, outside slums, less-affluent neighbor-
hoods of houses (vs. apartments) favored transmission. We showed that the cumulative
effects of temperature over the previous few weeks set the temporal limits for transmission
to occur, and that the outbreak was sparked by infected people arriving from regions with
ongoing outbreaks. Additionally, we implemented a statistical method to identify trans-
mission foci in real-time that improves targeting control interventions. Our results deepen
the understanding of dengue transmission as a result of social, physical, and biological
processes, and pose multiple opportunities for improving control of dengue and other
mosquito-borne viruses such as Zika, chikungunya, and yellow fever.

## Introduction

Dengue cases are steadily increasing worldwide since the 1950s, currently reaching ~390 mil-
lion new cases annually, of which ~96 million manifest clinically [1]. The main vector of den-
gue virus (DENV) is the highly urbanized mosquito *Aedes aegypti*, which also transmits other
viruses of public health importance, such as yellow fever, Zika, and chikungunya [2–4]. Grow-
ing urbanization, increasing human mobility at all scales, expansion of the geographic range of
*Ae. aegypti* after discontinuing eradication programs, and global warming are usually postu-
lated to underlie the rapid increase of cases [5]. These processes reflect specific modes of pro-
duction and social orders that shape the way people live and define who gets sick [6].

DENV transmission is strongly seasonal. In endemic regions, seasonality is considered to
be driven by a combination of the effects of rain, which increases the availability of artificial
breeding sites where immature *Ae. aegypti* develop, temperature, and the intrinsic dynamics of
host-parasite interaction [7,8]. In non-endemic regions, temperature is the major driver of sea-
sonality. Cool/cold winters virtually halt the development and activity of *Ae. aegypti* [9–11],
and the development of DENV within the vector, known as the extrinsic incubation period,
EIP [12,13]. Warm/hot summers allow the development of both *Ae. aegypti* and DENV. Rain
may also contribute to seasonality according to how many breeding sites depend on rain to get
filled [14]. Each season, each region is characterized by the predominant circulation of one of
the four distinct virus serotypes (DENV 1–4) [15,16]. Particularly in non-endemic regions,
DENV transmission relies on infected people coming from areas with DENV introducing the
virus each new season [17].

DENV transmission tends to be highly clustered spatially within a city or town [18–20].
The occurrence of clusters has been associated (mostly statistically) with various social and
environmental factors such as economic income, crowding, education level, water supply,
housing characteristics, human behavior, vegetation abundance, and *Ae. aegypti* presence/
abundance [21–25]. However, particularly for social factors discrepant effects have been
reported across different studies and areas [26–29]. Likewise, the spatial characteristics of clus-
ters of cases vary among study areas [19,20,30–33]. This variability likely denotes the diversity
of assessed settings and how different socioeconomical, political, and environmental settings
shape the biological processes underlying DENV transmission, emphasizing the need for a
local understanding of transmission.

Controlling DENV transmission focuses on vector suppression by targeting breeding sites
round the year, or by focal control during outbreaks around areas where transmission is

detected [34]. Focal control targets areas based on the residential address of detected (i.e., mostly symptomatic) cases. Although this does not focus on other possible (or actual) places where transmission occurs [33,35], it concentrates in hotspots of transmission. However, the criteria used for identifying these hotspots do not usually rely on rigorous statistical methods–although epidemiological methods have been developed and applied experimentally to dengue [36]–hindering optimal allocation of limited resources for vector control [37].

Seasonal DENV transmission in non-endemic regions has received limited attention [19,20,24,38–42]. A better understanding of transmission in these areas will shed light on several aspects of DENV transmission beyond non-endemic regions by enabling assessment of the initial phases of an outbreak when DENV begins circulating [17,43]. This is difficult in endemic areas where DENV has a virtually uninterrupted transmission. Likewise, there are limited analyses of local dynamics of transmission in large cities (i.e., with several million inhabitants), particularly, large cities in non-endemic regions [39]. Given ongoing global warming and the concomitant expansion of *Ae. aegypti* and of DENV transmission, the currently non-endemic regions are likely to become endemic rather soon [44]. Thus, it is of utmost importance to improve our understanding of transmission in these settings.

This study aims to describe the processes driving the spatial dynamics of the largest dengue outbreak until then in the non-endemic metropolis of Buenos Aires, Argentina, in 2016. It addresses i) how the outbreak began (characterizing the origin and timing of imported cases), ii) the temporal variation in transmission (assessed in terms of the effects of temperature and rain), iii) the spatial variation in transmission (statistically evaluating the relevance of different sociodemographic variables), iv) the spatial structure of transmission (characterizing the spatial scale of clustering), and v) how to identify hotspots in real-time while accounting for the heterogenous population distribution.

## Methods

### Ethics statement

All data on dengue cases were part of the information routinely collected for surveillance purposes by the local health authorities. All data were anonymized prior to analyses. No ethical approval was required.

### Study area

The Autonomous City of Buenos Aires (hereon, Buenos Aires) has a population of ~2.9 million in an area of 204 km$^2$ and is part of the largest urban conglomerate of South America which encompasses ~13 million people (29% of Argentina population). The city has vast areas dominated by apartment buildings and some industrial areas; ~24% of the dwellings are single-family houses (hereon, houses) and harbor ~30% of the inhabitants, the remainder living in apartments. A large proportion of houses are located in middle-class, low-rise residential areas usually with small gardens or courtyards and many with roof terraces. A portion of the most underprivileged people in the city live in slums (population, ~186,000; area, 9.3 km$^2$), characterized by very precarious housing, high population density, poor or no urban planning (i.e., limited access to running water and proper sewage), poor garbage collection services, and high violence, among other characteristics. The temperate climate of Buenos Aires has hot humid summers (average low, 19.3˚C; average high, 29.1˚C; mean rainfall, 384.4 mm) and rather cold winters (average low, 8.1˚C; average high, 16.2˚C). These temperatures allow *Ae. aegypti* development approximately from October to May [45]. DENV transmission in Argentina was first recorded in 1997, with minor seasonal transmission, until 2009 when the first major outbreak took place and the first autochthonous cases were registered in Buenos Aires

(240 confirmed cases, 600 suspected cases) [46]. In 2016 the hitherto largest outbreak in Argentina took place, with 41,207 confirmed autochthonous cases, mostly concentrated in Northeastern Argentina [47]. Since DENV introduction, local health authorities have implemented control measures intermittently, focusing on breeding sites year-round and focal control during outbreaks. In 2016, although no effectiveness was measured, focal control was very scarce with little chance of having any relevant effect on transmission.

### Data

Data on dengue cases during the 2016 season (end of 2015 to autumn 2016) were obtained from cases reported by healthcare facilities to the SNVS (National System of Health Vigilance, Ministry of Health of Buenos Aires City) and from medical records. These data included all cases for which dengue was considered a possible diagnosis. Following laboratory studies (PCR, NS1 antigen detection, and/or serology) or epidemiological assessment, cases were classified as confirmed (positive PCR or symptomatic with epidemiological link), probable (symptomatic positive to NS1 or IgM without epidemiological link), suspected (compatible symptoms without lab or epidemiological confirmation), or discarded (IgM negative or alternative diagnosis confirmed) [48]. Some cases included serotyping of DENV, particularly for initial cases. Health personnel registered the date of onset of symptoms, date of first medical visit, and travel history in the 15 days prior to onset of symptoms. A case was considered imported if the person had recently travelled to a region (either another country or province within Argentina) with an ongoing dengue outbreak at that time. Cases were georeferenced to their reported residential address, either street and number, intersection or slum name and block number. For cases with an address indicated by the name of a slum but with no street or block indication (14.4% of cases in slums) a random location within the slum was assigned to avoid concentrating those cases in a single point. Cases with incomplete or incongruent address (1.9% of confirmed and probable cases) could not be georeferenced and were considered in non-spatial analyses but not in spatial and spatiotemporal analyses. Confirmed cases represented 62.1% of cases assessed, while 5.3% were probable and 32.6% suspected ones. Only confirmed and probable cases were considered for the current analyses.

Data on daily mean temperature and daily rainfall were obtained from the National Meteorological Service for the weather station in Aeroparque, the local airport, which is located in the city and is representative of temperature and rainfall within the city. Demographic data at census tract level were obtained from the 2010 national survey [49]. Census tracts, the highest resolution available for census data, had a median area of 3.55 ha (90% interquantile range, 1.24–12.10 ha) and a median population of 786 (440–1,229). For the hierarchical modeling, census tracts were grouped into census fractions which had a median size of 39.4 ha (13.2–110.9 ha) and a median population of 7,994 (5,433–11,334). Large uninhabited areas (totaling 34.3 km$^2$) were manually identified based on satellite images [50] and knowledge of the terrain and excluded from analyses. Demographic and geographical information on slums was obtained from the City Housing Institute (https://vivienda.buenosaires.gob.ar/integracion/). The distinctive characteristics for defining slums are that they have no urban planning, building regulations are not followed, and most ownership is irregular [51].

Data on monthly inbound border crossings were obtained from the migration office of Argentina (Dirección Nacional de Migraciones). Data had a monthly resolution for each individual checkpoint with a bordering country with registered dengue transmission. Since it was not possible to get information on the origin of each person entering Argentina, only overland checkpoints were considered, assuming that incoming people had stayed in the neighboring country corresponding to each checkpoint. These checkpoints were 500–2000 km away from

Buenos Aires. Nevertheless, this traffic is informative of the international movement of inhabitants of Buenos Aires since the vast majority of people traveling to neighboring countries with DENV transmission tend to go overland through these checkpoints.

## Temporal variations: Effects of temperature and rainfall

In order to understand the overall temporal variations in DENV transmission, the weekly incidence rate ratio (IRR), calculated as the number of new cases during week $t$ divided by the number of new cases during week $t$ - 1, was assessed in relation to temperature and rainfall, regardless of the spatial dimension. Temperature governs *Ae. aegypti* and DENV development and its effects are a result of the specific time series of temperature. Therefore, we summarized the cumulative and delayed effect of temperature along time using a temperature-sensitive model of the basic reproduction number, $R_0(T)$ [52]. This model accounts for the effects of temperature on development, behavior, and mortality of *Ae. aegypti*, and on the EIP. By calculating the daily effects of temperature on these processes, we estimated the developmental history of the mosquitoes transmitting DENV on each given day. This enabled to account for the delay between hatching and adult biting and, therefore, to calculate the rainfall at the approximate time of hatching of each cohort and estimate $R_0(T)$ for each day accounting for the cumulative effects of recent temperatures. In this we considered the incubation period in humans [12,53]. The absolute value of $R_0$ indicates when transmission is sustainable but depends on additional elements (e.g., density of people, frequency of human-mosquito contact, housing, and vector control) not captured in this model. Therefore, since our goal was only to estimate temperature cumulative effects, $R_0(T)$ was rescaled from zero to one–which keeps $R_0(T) = 0$ (when no transmission is possible) unaltered–. Further details in S1 Methods.

## Spatial variations: Hierarchical modeling of demographic variables

Assessing which variables are–statistically–related with the spatial distribution of autochthonous cases requires to account for the history of cases in each part of the city, even when the assessed variables may not change temporally. For instance, a very favorable area for transmission (plenty of *Ae. aegypti* and opportunities for human-mosquito contact) may show no cases just because no infected people introduced DENV in that specific area. This is particularly relevant when a low proportion of all inhabitants becomes infected as in the 2016 outbreak. Additionally, an unknown–but estimable–fraction of cases goes unregistered [54].

A hierarchical spatiotemporal model was built to assess the effects of demographic covariates on DENV transmission outside of slums while accounting for the temporal autocorrelation of cases. Slums were excluded as their specific social and environmental characteristics are distinctively associated with overall poor health. Excluding slums allowed better understanding of which factors were related to DENV transmission in the rest of the city. The spatial unit, $i$, was the census fraction; the time unit, $t$, was a week. Due to the likely possibility of overdispersion in data, the number of cases was assumed to follow a negative binomial distribution with mean $\mu_{it}$ and success probability $\phi$. The mean, $\mu_{it}$, was modeled as the linear combination of: i) the joint effects of the demographic covariates $v_j$ given the number of previous cases, $y_{i,t-1}$, occurring in census fraction $i$ on the previous week, $t$ - 1, ii) the contribution of neighboring cases, $z_{i,t-1}$, on the previous week in fractions immediately adjacent to $i$, iii) the contribution of overall cases, $\Sigma_i\, y_{i,t-1}$, in all census fractions of the city on the previous week, and iv) an overall constant term $\gamma_0$. Overall cases were considered because it was hypothesized that a higher overall incidence in the city implied higher chances that any inhabitant got infected somewhere in

the city regardless of its neighborhood suitability.

$$Y_{it} \sim \text{Binomial}(y_{it}, \alpha)$$

$$y_{it} \sim \text{NegBinomial}(\mu_{it}(1 - \phi)\phi^{-1}, \phi)$$

$$\mu_{it} = y_{i,t-1}\exp\left(\beta_0 + \sum_{j=1}^{V} \beta_j v_{ji}\right) + \gamma_0 + \gamma_1 \sum_{i=1}^{N} y_{i,t-1} + \gamma_2 z_{i,t-1}$$

The number of autochthonous DENV cases resulting from the negative binomial distribution became the number of trials of a binomial distribution. This was a way of accounting for missing observations of cases. Thus, $\alpha$ would be the probability of observing/detecting a DENV case in a week given that it was present in that week (complete formulation of the model in S1 Methods).

For each census fraction $i$ the considered covariates were the fraction of dwellings that were houses (as opposed to apartments), the fraction of dwellings with poor (as opposed to good) building quality, the fraction of population having no further than high school studies, the fraction of unemployed population, and the fraction of homes with ≥2 people per room (as indicative of overcrowding) (Fig 1). These covariates were centered and standardized to facilitate comparison between their effects. The chosen prior distributions were weakly informative and ensured that $\mu_{it} \geq 0$ and $0 < \alpha \leq 1$ (S1 Methods).

The model was fitted to observed data using Markov Chain Monte Carlo sampling using the Nimble package [55,56] in R [57]. Briefly, four chains were run, each with 90,000 total iterations, 9,000 burn-in iterations, and a thinning interval of 3. The effect of each covariate $v_j$ was evaluated according to which fraction $f$ of the posterior samples of its coefficient $\beta_j$ had the same sign as the mean of the posterior. Further details can be found in the S1 Methods.

## Spatial structure and foci detection: Spatiotemporal analysis

To describe the overall spatial distribution of cases, global spatial analyses were performed using the mark-connection function [58]. The mark-connection function evaluates the spatial distribution of qualitatively marked points within a ring of given radius and width. This contrasts with more popular statistics (such as Ripley's $K$- and $L$-functions) that evaluate distribution within a disc of a given radius. The disc implies that effects at shorter distances obscure effects at larger distances [59]. Considering a ring avoids this issue; thus, the mark-connection function enables characterization of patterns by assessing independently for each distance the contribution to the observed spatial pattern.

In turn, local spatiotemporal analyses were performed to identify areas of statistically significant spatiotemporal clustering. This, applied in real-time, would serve as a tool to prioritize and target control efforts during an outbreak. These analyses were performed weekly using the local $K$-function calculated for each point on a uniform grid of 100 m cells [60]. The $K$-function enabled assessment of the whole area surrounding each reference point within a given radius, instead of only a ring as with the mark-connection function. Likewise, the time window considered for the local analysis spanned from week $t$–$\tau$ to week $t$, where $\tau$ is analogous to the spatial radius [61]. Such cumulative function was chosen, instead of a ring function, because the global analyses showed an almost monotonical decrease of clustering with distance and the interest was in detecting clusters rather than characterizing them. To define what radius and time window to use, the average distance and time of clustering due to local transmission at a weekly base was characterized performing a global spatiotemporal analysis using the mark-

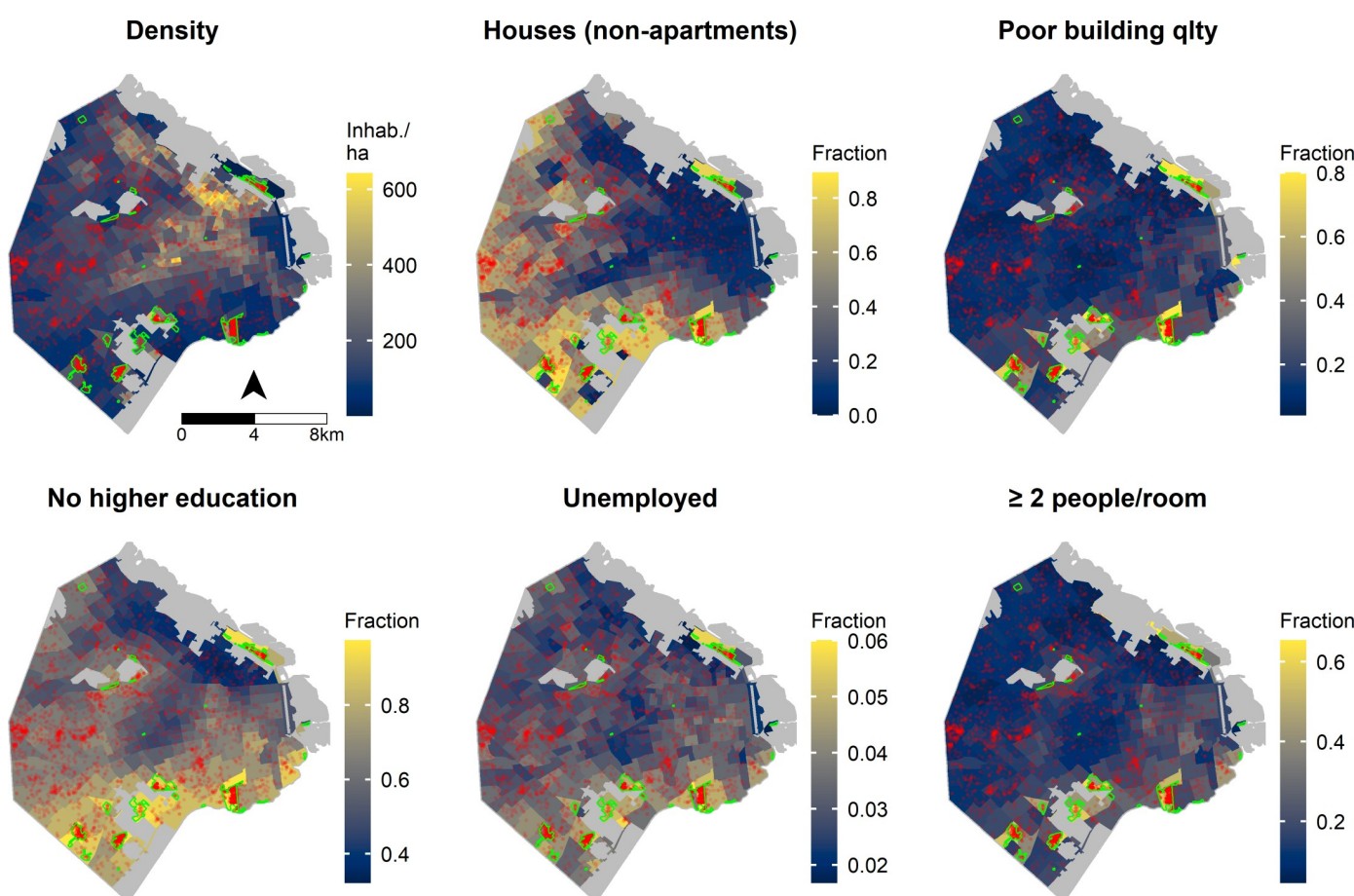

**Fig 1. Distribution of inhabitant density, demographic covariates, and dengue autochthonous cases in Buenos Aires.** Covariates are quantified as the fraction (or proportion) of dwellings or people–depending on the covariate–in each census fraction that correspond to the considered category. These five covariates were included in the hierarchical model. All autochthonous cases occurring during the outbreak are displayed as red points. Grey shaded areas are uninhabited. Green polygons show the limits of slums. The spatial unit displayed is the census fraction. Map base layer downloaded from the open-data repository of the Government of the City of Buenos Aires (https://data.buenosaires.gob.ar/dataset/informacion-censal-por-radio).

connection function. The temporal dimension was incorporated by considering all cases within a given space-time ring analogous to the spatial ring. The initial phase of the outbreak would provide the best estimates of typical clustering distance since at that time most clusters were still small, more likely reflecting the transmission pattern from a few cases.

For both global and local analyses, the observed patterns were compared to random patterns generated by random labeling (labels being "infected" and "not infected") among the fixed location of the homes of all inhabitants in the city [59]. Analyses only evaluated the location of infected individuals (either observed or simulated). The random labeling ensured that the simulated random patterns accounted for the underlying population distribution, thus, distinguishing the observed clustering process from artifacts arising from the heterogeneous population distribution. The locations of the inhabitant homes were roughly estimated by uniformly distributing the number of inhabitants of each census tract among the area of the census tract. Excluding large uninhabited areas helped reducing distortions in population spatial distribution. For each analysis 500 simulations were performed and used for building 95% confidence envelopes. In the case of the local analysis, cells with a *K*-function value above the confidence envelope for that cell was considered as having clustered cases within the radius

and time window specified. All spatial analyses were performed using Spatstat package version 1.64–1 [62] for R.

## Results

### Overall dynamics

Out of a total of 5,202 reported dengue cases in Buenos Aires, 422 (8.1%) were allegedly imported cases whereas the remaining 4,780 cases were considered autochthonous. The outbreak began approximately at the turn of the year with an increasing number of reported imported cases (Fig 2). Concomitantly, autochthonous cases increased, exhibiting a sharp growth from mid-February to mid-March, reaching steady numbers for ~4 weeks, until the beginning of April. By mid-May, the outbreak was over (Fig 2). Out of the 5,104 georeferenced cases, 1,682 (33.0%) were located in slums, with 1,538 (32.8%) autochthonous cases (Fig A in S1 Results). The dynamics of new cases was similar between slums and the rest of the city (Fig B in S1 Results). The proportion of imported cases in slums and outside slums did not differ significantly (Fisher test, $P = 0.32$). DENV per-capita incidence in slums was higher than outside slums (overall 7.3 times greater), even after stratifying by the different demographic covariates considered for the hierarchical model (Fig C in S1 Results).

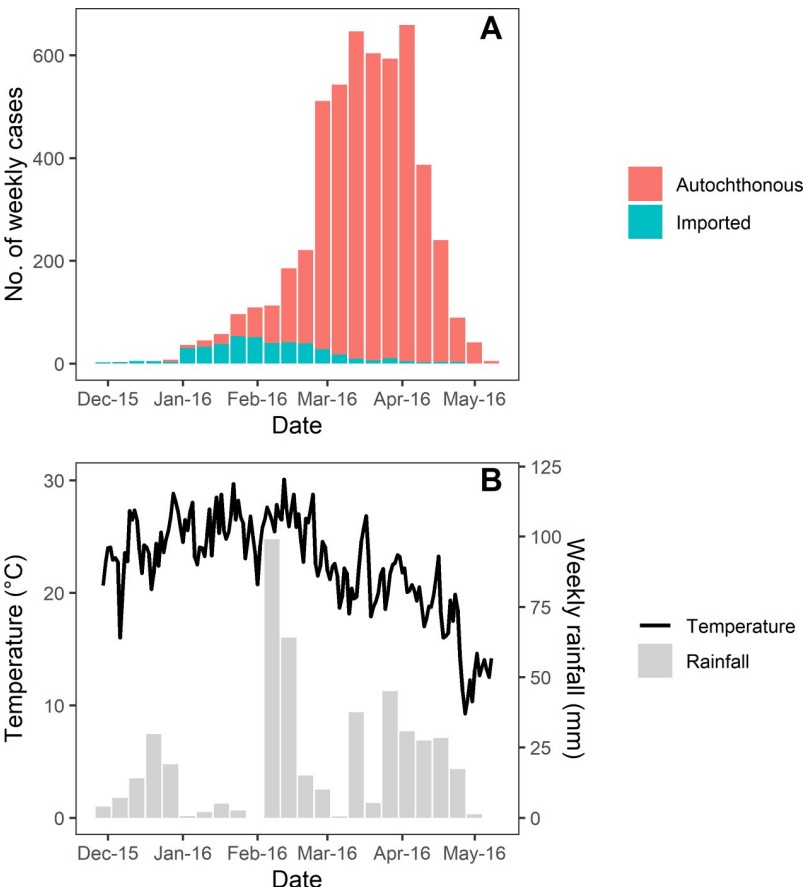

**Fig 2. Overall dynamics of dengue transmission, temperature, and rainfall in Buenos Aires during the 2016 outbreak.** A) Weekly cases of dengue according to date of onset of symptoms and origin of infection. B) Daily mean temperature (black line) and weekly rainfall (grey bars).

## Onset of transmission: Imported cases

The major origin registered for imported cases was Paraguay (60.4%), followed by Northeastern Argentina (21.1%), Brazil (7.6%), northern South America and The Caribbean (3.1%), and Northwestern Argentina (2.6%). Most imported cases in slums came from Paraguay (83.3%) and Northeastern Argentina (9.0%), whereas outside slums these origins represented 47.2 and 27.9% of importations, respectively (Table 1). People incoming overland from Paraguay represented 64.3% (1.2 million people in January 2016) of all border crossings with neighboring countries registering dengue transmission, followed by people incoming from Brazil (19.2%, 353,662 people) (Fig D in S1 Results).

A total of 319 (6.1%) of the confirmed cases was positively serotyped. DENV-1 was detected in 97.8% of the autochthonous and in 94.7% of the imported serotyped cases (Table 2). The remaining serotyped cases were DENV-2 and DENV-4. Two of these cases were located in different slums, whereas the others were located elsewhere, away from other cases registered with their serotype. Serotyping was relatively more frequent for imported than for autochthonous cases, as most serotyping was concentrated during the beginning (January and February) of the outbreak.

## Temporal variations: Effects of temperature and rainfall

Taking the normalized $R_0(T)$ as a summary of the cumulative effects of temperature, the weekly IRR followed closely $R_0(T)$ dynamics (Kendall correlation coefficients: median = 0.68, 95% CI = 0.61–0.75, $P < 0.05$), evidencing the strong effects of temperature on DENV transmission (Fig 3). In fact, $R_0(T)$ exhibited very low values until mid-December, and reached relatively high values at the beginning of January (Fig 3B), precisely when the outbreak began. At the end of February, after the sharpest increase in autochthonous cases (i.e., the maximum IRR), $R_0(T)$ fell steadily, concomitantly with the approximate plateau (IRR $\approx$ 1) and later decrease (IRR < 1) displayed by autochthonous cases (Figs 2A, 3A and 3B). In contrast, the weekly rainfall at the estimated time of egg hatching of the *Ae. aegypti* supposedly involved in transmission did not show a clear correspondence with transmission dynamics (Kendall correlation coefficients: median = -0.18, 95% CI = -0.41 – -0.04, $P > 0.05$) (Fig 3A and 3C).

## Spatial variations: Hierarchical modeling of demographic variables

The hierarchical modeling identified the fraction of houses per census fraction and the fraction of inhabitants with no formal studies beyond high school as the most influential covariates on

**Table 1. Number of imported dengue cases in Buenos Aires according to likely origin of infection and residential address located in or outside slums.**

| Origin | Outside slums | In slums | Total |
|---|---|---|---|
| Paraguay | 120 (83.3%) | 125 (47.2%) | 255 (60.4%) |
| NE Argentina | 13 (9.0%) | 74 (27.9%) | 89 (21.1%) |
| Brazil | 0 (0.0%) | 32 (12.1%) | 32 (7.6%) |
| Caribbean & Central America | 0 (0.0%) | 13 (4.9%) | 13 (3.1%) |
| NW Argentina | 6 (4.2%) | 4 (1.5%) | 11 (2.6%) |
| Peru | 1 (0.7%) | 7 (2.6%) | 8 (1.9%) |
| Bolivia | 4 (2.8%) | 3 (1.1%) | 7 (1.7%) |
| N S. America | 0 (0.0%) | 5 (1.9%) | 5 (1.2%) |
| Asia | 0 (0.0%) | 2 (0.8%) | 2 (0.5%) |
| Total | 144 (100.0%) | 265 (100.0%) | 422 (100.0%) |

**Table 2. Number of dengue cases in Buenos Aires according to serotype and likely origin of infection.**

| | | Serotype | | | | |
|---|---|---|---|---|---|---|
| Origin | | DENV-1 | DENV-2 | DENV-4 | *Not serotyped* | Total |
| Autochthonous | | 179 | 2 | 2 | 4,597 | 4,780 |
| Imported | Paraguay | 81 | 1 | 4 | 169 | 255 |
| | NE Argentina | 31 | - | - | 58 | 89 |
| | Brazil | 9 | - | - | 23 | 32 |
| | Caribbean & Central America | 3 | - | - | 10 | 13 |
| | NW Argentina | 3 | - | - | 8 | 11 |
| | Peru | 2 | 1 | - | 5 | 8 |
| | Bolivia | - | - | - | 7 | 7 |
| | N S. America | - | 1 | - | 4 | 5 |
| | Asia | - | - | - | 2 | 2 |
| Total | | 308 | 5 | 6 | 4,883 | 5,202 |

DENV transmission outside slums, both with a positive effect on the number of cases. Unemployment showed a weaker effect with 95% of its posterior being negative. The remaining covariates–building quality and overcrowding–showed effects not clearly different from zero, with $f \leq 0.87$ (Table 3). The total number of autochthonous cases in the city on the previous week had a small contribution ($\gamma_1 = 1.1 \times 10^{-3}$ per case), whereas the effect of adjacent neighbors was negligible ($\gamma_2 = 5.1 \times 10^{-4}$ per case). The hierarchical model estimated that the probability $\alpha$ of observing an autochthonous case in a week given that it was present averaged 0.16 (95% CI: 0.11–0.24). The fitted value of the success probability $\phi$ of the negative binomial distribution showed that a certain degree of overdispersion remained in the model data. Model residuals distributed randomly in space, evidencing that the model captured the spatial structure present in the observed number of cases (Fig E in S1 Results). Additionally, 98.3% of the observed values fell within the 80% interquantile range of the distribution of the predictive posterior. All these results held when considering only confirmed cases as well as a Poisson instead of a negative binomial distribution (Tables A-C in S1 Results).

### Spatial structure: Global spatial analyses

The overall spatial distribution of imported cases was clustered at <600 m (Fig 4A). Excluding the 265 imported cases in slums showed that imported cases in the rest of the city (144 cases) were not clustered between them. In turn, autochthonous cases were clustered in and outside slums (and overall) up to 600 m and 900 m, respectively (Fig 4D and 4F). The spatial association between all imported and all autochthonous showed clustering up to ~800 m (Fig 4G) but this owed to the contribution of the clustering in slums (Fig 4I). When excluding cases in slums, autochthonous cases were not clustered around imported cases (Fig 4H).

The global spatiotemporal analysis performed weekly for different space-time rings showed that cases outside slums were not statistically clustered until the end of February, precisely when transmission rose sharply (Fig F in S1 Results). Until then, cases were clustered mostly at 250–400 m within the current week as well as with respect to cases one week earlier; clustering was less often observed for time lags of two or three weeks. During the following weeks (March and first half of April), clustering distance increased with respect to cases in the 1–3 previous weeks (as clusters consolidated and increased in size) ranging 500–800 m, while clustering within the same week became rather negligible.

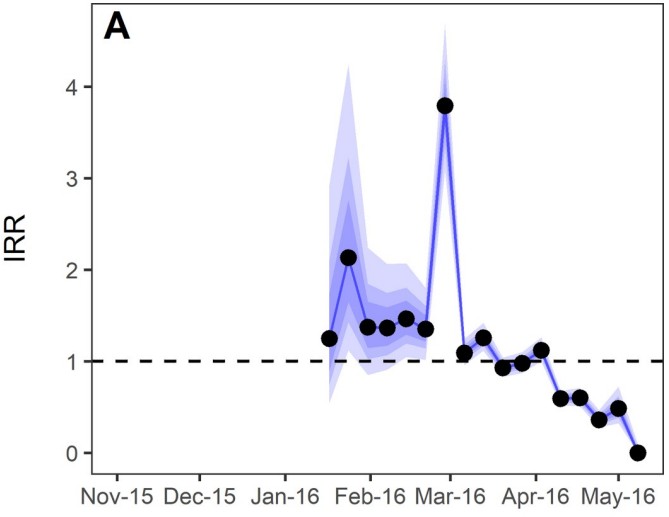

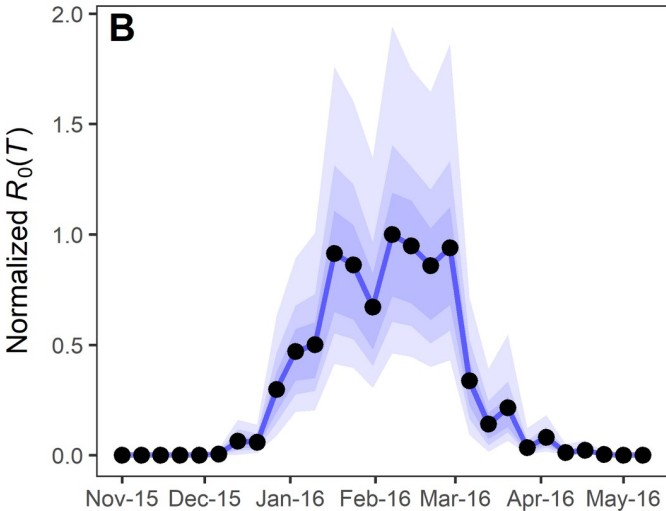

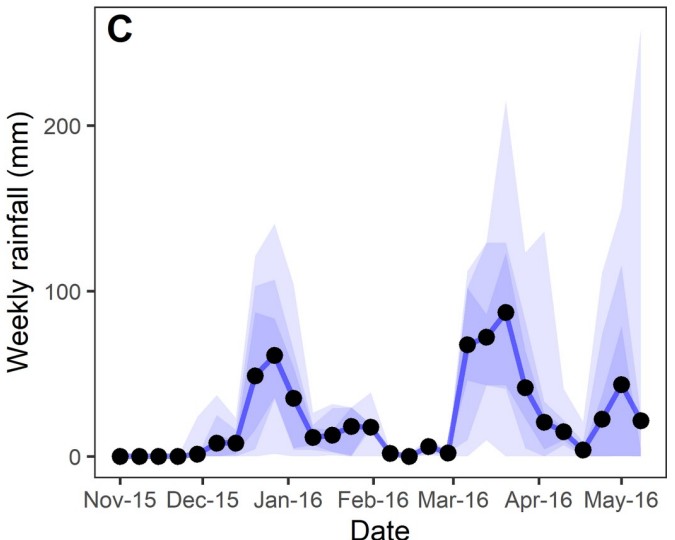

**Fig 3. Dengue transmission follows closely $R_0(T)$ but not rainfall in Buenos Aires during the 2016 outbreak.** A) Weekly incidence rate ratio (IRR) of autochthonous cases; the dashed line indicates the IRR value of steady weekly cases (IRR = 1). B) Normalized temperature-dependent basic reproduction number, $R_0(T)$, as a summary of the cumulative effects of temperature along time on transmission. C) Weekly rainfall at the estimated time of egg hatching of *Ae. aegypti* that were putatively involved in DENV transmission on each given date. Black dots indicate the value corresponding to the beginning of the epidemiological week. Shaded areas indicate the 50, 75, and 95% confidence intervals.

## Foci detection: Local spatiotemporal analysis

Based on the global spatiotemporal analysis (Fig F in S1 Results) we considered a radius of 300 m and a time window encompassing only the current week for the local spatiotemporal analysis. This identified many of the intuitively recognized patterns, particularly those in less populated areas (Fig 5). Overall, 53.1% of cases belonged to clusters. Within slums, 91.3% of cases and 80.0% of the population were in clusters; outside slums, 34.9% of cases and 20.4% of the population were in clusters. Exposition to DENV transmission was also longer in slums than outside slums: 41.9% of the population in slums against 0.3% outside slums were in clusters for ≥10 weeks, whereas for 1–9 weeks 38.1% in slums and 20.2% outside slums were in clusters (Fig G in S1 Results). Increasing the radius and/or time window assessed translated in only slightly more cases but much more population included in clusters (Fig H in S1 Results). Increasing the time window slightly increased the size and duration of clusters (Fig H in S1 Results), mainly because of the "memory" that widening the time window implied, as areas on a given week could be identified as clusters based on cases on the previous weeks, even without cases on the current week (Fig I in S1 Results). This became particularly evident and eventually misleading during the end of the outbreak.

## Discussion

We depicted in detail the development of a major dengue outbreak in a non-endemic subtropical metropolis. We found that most people became infected in or near their home, particularly in slums and secondarily in low-rise neighborhoods of houses, and that transmission closely tracked the accumulated effects of temperature over time, while rainfall played a negligible role. Imported cases arriving when temperature was most favorable sparked local

**Table 3. Hierarchical spatiotemporal modeling of dengue transmission during the 2016 outbreak in Buenos Aires.**

| Variable | Mean | Q2.5% | Q97.5% | f | r-hat | N eff |
|---|---|---|---|---|---|---|
| Houses | 0.33 | 0.20 | 0.50 | 1.00 | 1.00 | 1,023 |
| No higher studies | 0.27 | -0.07 | 0.58 | 0.95 | 1.00 | 1,075 |
| Good building qlty. | 0.18 | -0.14 | 0.51 | 0.87 | 1.00 | 860 |
| Unemployed | -0.11 | -0.33 | 0.09 | 0.85 | 1.00 | 3,119 |
| Overcrowding | -0.03 | -0.35 | 0.30 | 0.59 | 1.00 | 730 |
| *Constant* ($\gamma_0$) | 0.107 | 0.014 | 0.217 | 1.00 | 1.00 | 2,491 |
| *City cases* ($\gamma_1$) | 0.0011 | 0.0008 | 0.0014 | 1.00 | 1.00 | 1,609 |
| *Neighbors* ($\gamma_2$) | 0.00051 | 0.00001 | 0.00188 | 1.00 | 1.00 | 23,971 |
| *Detection probability* ($\alpha$) | 0.16 | 0.11 | 0.24 | 1.00 | 1.05 | 324 |
| *Prob. NegBin* ($\phi$) | 0.51 | 0.34 | 0.74 | 1.00 | 1.00 | 715 |

The effects of the demographic covariates and of neighboring and citywide cases are displayed. The mean and 95% interquantile range of the posterior distribution of the corresponding model parameters are shown, together with the fraction *f* of the posterior with the same sign as the mean, r-hat as indicative of convergence and 'N eff', the effective sample size.

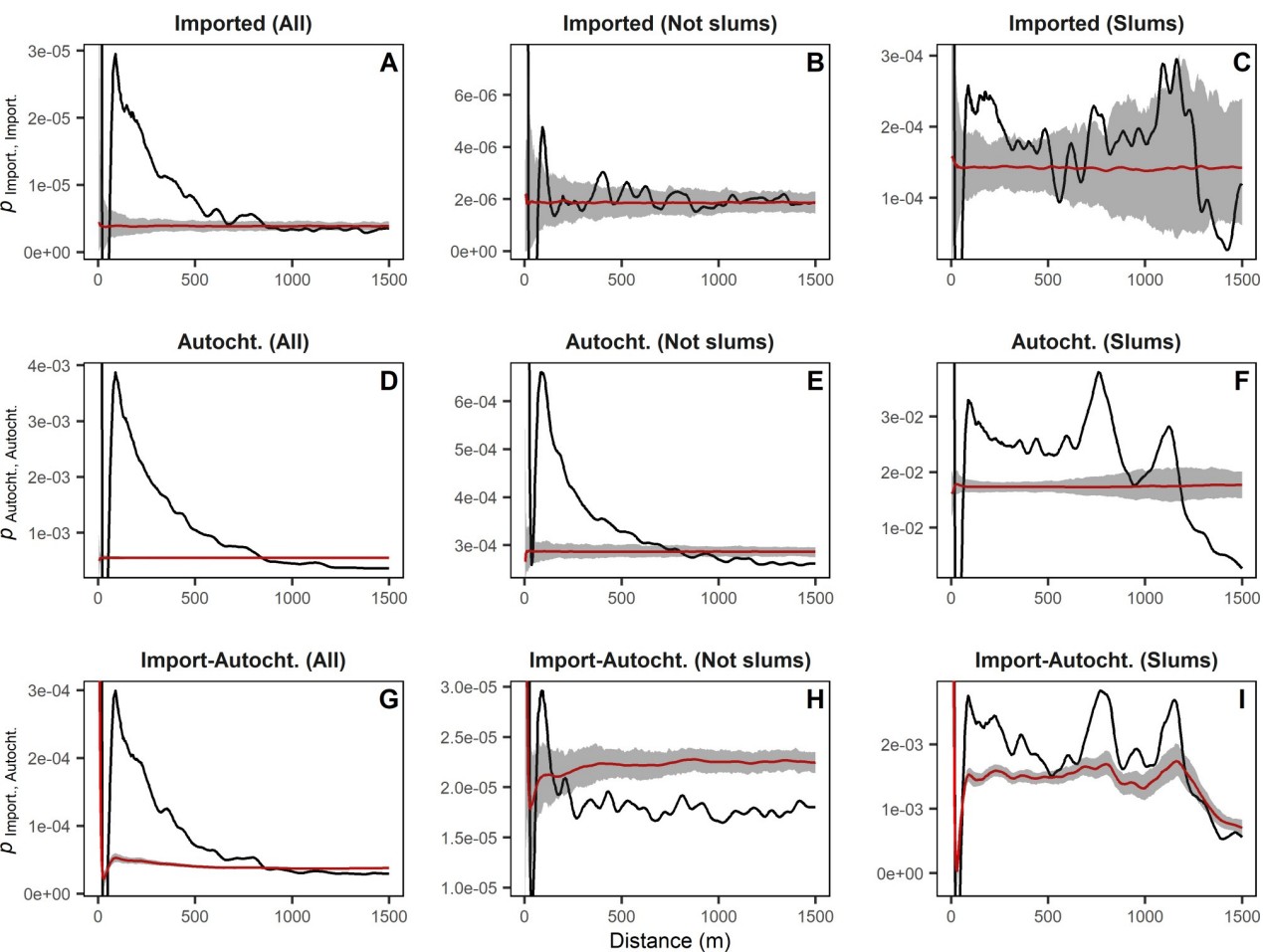

**Fig 4. Imported dengue cases clustered only in slums, whereas autochthonous cases clustered citywide.** The mark-connection function, *p*, assessed the spatial distribution between the groups of selected cases in Buenos Aires. A, B and C) Imported cases. D, E and F) Autochthonous cases. G, H and I) Autochthonous cases according to imported cases. A, D and G) Cases in all the city. B, E and H) Only cases outside slums. C, F and I) Only cases in slums. The black line shows the values for the observed pattern. The red line shows the mean value for the random patterns. The shading represents the 95% confidence interval for the random patterns. Random patterns were generated by random labeling among the inhabitants of the city. Observed values above 95% confidence interval denote clustered distribution.

transmission. We traced back these imported cases establishing that the main source of DENV was an ongoing outbreak in the country with which Argentina has the highest flow of people. Using a hierarchical Bayesian model, we could separate the effect of different covariates and other factors affecting transmission from the imperfect detection of DENV cases. This not only allowed for unbiased estimates of coefficients but also gave us an estimate of the fraction of undetected DENV cases, providing a method complementary to more resource demanding approaches. Lastly, we implemented a statistically rigorous method for detecting transmission foci in real-time while accounting for the heterogeneous population distribution. By characterizing DENV transmission by *Ae. aegypti*, our results also shed light on the eventual transmission of the rapidly expanding Zika and chikungunya viruses.

Transmission occurred 7.3 times more frequently in slums compared to the rest of the city, most population in slums were in transmission foci at some point, and most foci lasted much longer in slums than outside. Serious and systemic deficiencies in garbage collection, house infrastructure (e.g., proper covering of water tanks, if any), water provision, public and private

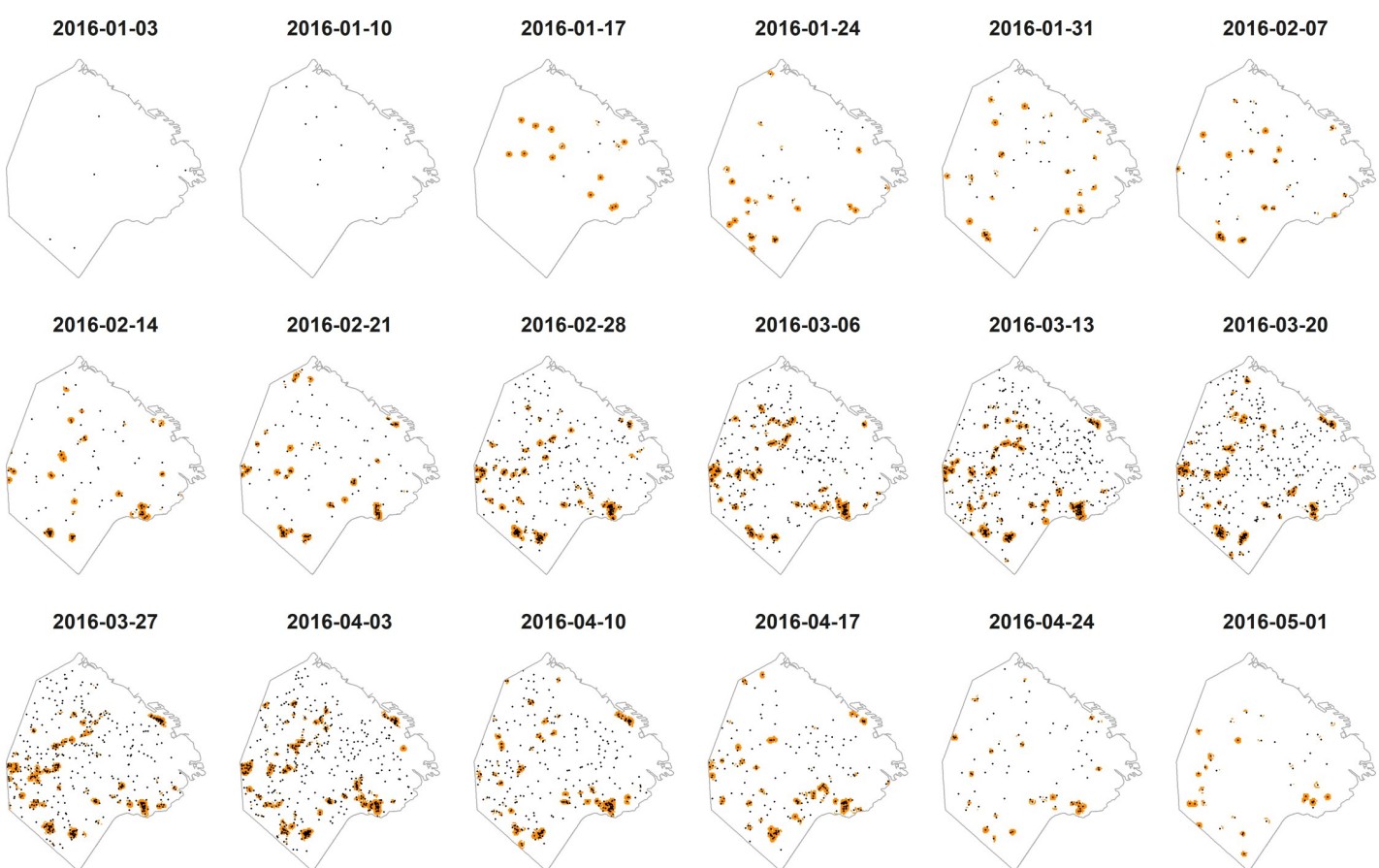

**Fig 5. Weekly local spatiotemporal analysis identified multiple dengue transmission foci in Buenos Aires.** Foci are indicated by orange areas. Autochthonous cases are displayed as black points. The displayed dates indicate the beginning of the corresponding week. The time window considered encompassed only the current week (i.e., $\tau = 0$), with a radius of 300 m. Map base layer downloaded from the open-data repository of the Government of the City of Buenos Aires (https://data.buenosaires.gob.ar/dataset/informacion-censal-por-radio).

cleanliness, poor roof drains, and informal dumps generate suitable habitats for *Ae. aegypti* immature development [63]. All this, together with overcrowding, social violence, and low political visibility, hinders DENV control interventions, configuring a worrisome scenario for health. Outside slums, the spatiotemporal hierarchical analysis showed that transmission mostly occurred among those with poor formal education–probably reflecting poorer (or worse off) settings that favor DENV transmission [25,28]–and in neighborhoods where houses prevailed. These neighborhoods not only help human-mosquito contact [64], but also have gardens and courtyards (sometimes with litter), unlidded water tanks, wasteland, and abandoned houses that can commonly propitiate *Ae. aegypti* development. The different sectors of the city pose very different challenges for effective control of *Ae. aegypti* and DENV transmission. High spatial and temporal resolution data on *Ae. aegypti* abundance would provide a more solid characterization of the high transmission areas, explaining how the sociodemographic characteristics link to the biophysical conditions that favor transmission. This would provide essential information to design and target control interventions.

Seasonality characterizes DENV transmission worldwide, although the precise drivers vary geographically together with climate and human behavior [7,65]. Our analysis of the cumulative effects of temperature over time on DENV transmission are consistent with the decisive role temperature has on allowing transmission to occur [52,66]. This warns about the

imminent consequences of global warming on increasing the duration and intensity of outbreaks while expanding the limits for more sustained and, eventually, endemic DENV transmission [44,67]. A more thorough comparison, possibly a hierarchical modeling, encompassing several more outbreaks is essential to establish how temperature conditions transmission at these latitudes. Additionally, the parameters taken from the literature [52] defining temperature effects on *Ae. aegypti* development and mortality–and, eventually, on DENV transmission–could be revisited and adjusted to the local mosquito strains, since adaptations to colder climates were recently found in *Ae. aegypti* in our region and in other subtropical/temperate regions [68–70]. In turn, rainfall, temporally displaced as to account for the delay between *Ae. aegypti* hatching and DENV transmission, showed no relationship with transmission in our case. Reports for other regions [71] and models for Buenos Aires [72] suggest a positive effect of rain on *Ae. aegypti* abundance. These discrepancies could reflect differences between regions in the fraction of breeding sites relying on rainwater.

In non-endemic regions, dengue outbreaks are sparked by imported cases [17,42]. In the Buenos Aires outbreak, most imported cases came from Paraguay–where a large dengue outbreak was taking place [73]–and adjacent Northeastern Argentina–also registering numerous cases [47]–. Dengue outbreaks in Paraguay would represent a high risk for dengue occurrence in Buenos Aires (and likely other places in Argentina) as: i) ~35% of the foreign population living in Buenos Aires comes from Paraguay [49]; ii) these communities are highly concentrated in areas favorable for transmission, such as slums [49]; iii) most overland border crossing in Argentina is to and from Paraguay, and iv) border crossing peaks during summer, when DENV thrives in Paraguay and favorable conditions for transmission are found in Buenos Aires. Further research on the joint time series of dengue cases in neighboring countries and Argentina, accounting for human mobility, would help explain how the epidemiology in neighboring countries influences and could be used to predict DENV transmission in Argentina [74]. Meanwhile, the current results provide solid evidence to warn about likely outbreaks in Buenos Aires–and Argentina–according to regional epidemiology, allowing for enhanced early detection of imported cases, particularly in high-risk areas of the city.

At shorter distances, people's daily displacements within the scope of the city carry DENV to other neighborhoods [35,75,76], giving rise to autochthonous cases observed city wide, even in the absence of previous nearby cases. Our results showed DENV transmission to be highly focalized, with early clusters of 250–400 m in radius, and developed ones reaching on average 500–800 m, as revealed by the global spatial and spatiotemporal analyses. These clustering distances suggest that very local displacements of people underlie the observed gradual spatial growth of clusters [20,75–77], as *Ae. aegypti* adults usually move <50 m away, exceptionally reaching 300–400 m [78]. Similar clustering distances have been found in Cairns, Australia, and in Porto Alegre, Brazil [19,20]. The clustering time window of 1–2 weeks for the first detected clusters is in the order of the EIP duration (10–15 days in Buenos Aires), supporting the idea that observed clusters are due to local transmission. Altogether, the strong and persistent spatiotemporal clustering demonstrates that a major part of transmission occurs at home or in its vicinity, as noted elsewhere [19,76]. Nevertheless, transmission beyond the vicinity of the residence is an important phenomenon that needs to be considered–for instance, by contact tracing or statistically reconstructing transmission chains [20,35].

DENV outbreaks tend to have one predominant circulating serotype [16]. In our case, DENV-1 was the predominant serotype, in imported as well as in autochthonous cases. This serotype was the predominant in Paraguay, Brazil and the rest of South America during the 2016 season [73,79]. DENV-2 and 4, found in 2.5% of serotyped imported cases in Buenos Aires, were also circulating, with lower frequency, in the countries of origin [73,79]. The only four autochthonous cases diagnosed with DENV-2 or 4 could have become infected from the

minority imported cases with these serotypes or could have been misclassified imported cases. Serotyping more cases during the outbreak could provide more solid evidence of the origin of DENV as well as the patterns of local transmission.

The probability of detecting a dengue case encompasses not only issues in diagnosing patients that arrive to the health care facilities but also the existence of asymptomatic and sub-clinical cases who will not seek medical assistance. These asymptomatic and sub-clinical cases contribute to transmission in a similar manner as symptomatic cases or even more [54]. The hierarchical modeling estimated that only ~16% of cases were detected in Buenos Aires. This agrees with levels of underreporting and asymptomatic cases for other regions, widely ranging 3–20 times the detected cases (i.e., a detection probability of 0.05–0.3) [1,80–82]. The current results provide a local estimate for Buenos Aires and show the usefulness of hierarchical modeling in estimating underreporting and asymptomatic cases, complementing more direct but demanding methods such as massive serology or enhanced surveillance.

Foci detection during an outbreak helps to identify and focus control measures. Although most transmission appears to be fueled by asymptomatic/unregistered cases [54], the conventional surveillance systems detect mostly symptomatic cases. This makes early response to incipient transmission difficult to attain. However, as cases increase, detected (symptomatic) cases keep some proportionality with undetected/asymptomatic cases. Thus, detected cases enable identifying areas where transmission is important, allowing to guide control efforts to major transmission foci. We provide a statistically rigorous method for identifying these foci, accounting for population distribution. In Buenos Aires, it proved to be particularly helpful in identifying clusters of cases in the vast areas outside the already known foci. The currently proposed analysis relies on overall abundance of cases; it is helpful for detecting major areas of transmission but is blind to small chains of transmission, particularly in densely populated areas. Reconstruction of transmission chains, either statistically or by contact tracing, would help detect these small or incipient chains of transmission [20,54]. Even if contact tracing may not be fully feasible at a control scale in real-time in Buenos Aires, it would provide a key understanding on DENV transmission in this kind of cities.

The available data for this study brings about several limitations. Our data, as most data regularly collected by the health system, is passive, depending on those seeking health attention. This increases the number of unregistered cases (even of symptomatic and sub-clinical ones) [83]. Enhanced surveillance, if feasible at all in a city like Buenos Aires, would increase case detection. Also, only the residential address is registered, with no information of contact tracing or, more generally, mobility that would provide a more comprehensive understanding of where transmission takes place [35,77,84,85]. Thus, the analyses can only evaluate the hypothesis of transmission at or nearby home. Nevertheless, significant clustering of home locations of cases supports the importance of the residential address in transmission. The health system, usually focused on sick people, has a very partial account of discarded cases, something that would, otherwise, allow for a more rigorous spatial analyses using a case-control design. We lack entomological data that would also contribute to a more thorough picture of the transmission settings within the city. However, obtaining precise and extensive data on *Ae. aegypti* abundance in a 3-million inhabitant city is, at least, challenging. Lastly, the demographic data used, on which the spatial analyses rely, could be outdated six years after the census. However, the low rate of population growth (<0.3% per year) [86] in Buenos Aires suggests no major biases in the results. These limitations are common to many studies using (passive) health system data. It would prove helpful to find ways to obtain better, more detailed, and updated data to gain a more precise understanding of dengue transmission and to achieve a more effective control of transmission.

The steadily increasing transmission of DENV (and other arboviruses) worldwide evidences that controlling transmission is far from effective, and big cities like Buenos Aires are no exception [34]. Those in poverty–mostly in slums, in our case–are usually the weakest link. Besides delving into what particular conditions or behaviors expose them to higher DENV transmission, we should remain aware that these are only manifestations of entrenched socioeconomic processes that need to be addressed as part of a strategy beyond dengue so slums to stop being slums. Outside slums, our results provide evidence of sociodemographic characteristics associated with DENV transmission that allows stratifying risk to better target preventive and reactive interventions [87,88]. Interventions during an outbreak can be further focused in real-time by means of the local spatiotemporal analysis we implemented. This approach is additionally supported by our finding that in Buenos Aires–as also reported in several other cities [19,20,76,77]–most transmission occurred at or nearby the household, although this same finding poses that almost half of cases do not get infected near their alleged home or there is a considerable mis- or under-registration. Our results underscore the importance of rapidly and effectively identifying imported cases–which are particularly likely when some neighboring countries have ongoing outbreaks–early during the outbreak and promptly blocking transmission in the vicinity of their address in the city. All these findings and tools may prove useful for other subtropical/temperate cities and for chikungunya and Zika viruses, as these are transmitted by the same vector and follow similar dynamics as DENV [88]. Nevertheless, applying this knowledge relies on the availability of adequate resources, something often far from guaranteed in these settings.

## Supporting information

**S1 Methods. Further details on methodology.**
(PDF)

**S1 Results. Tables and figures of supplementary results.** Table A in S1 Results. Hierarchical spatiotemporal modeling of confirmed cases using a negative binomial distribution during the 2016 outbreak in Buenos Aires. Table B in S1 Results. Hierarchical spatiotemporal modeling of confirmed cases using a Poisson distribution during the 2016 outbreak in Buenos Aires. Table C in S1 Results. Hierarchical spatiotemporal modeling of confirmed and probable cases using a Poisson distribution during the 2016 outbreak in Buenos Aires. Fig A in S1 Results. All confirmed and probable dengue cases during the 2016 outbreak in Buenos Aires according to origin of infection. Fig B in S1 Results. Weekly cases of dengue in Buenos Aires during the 2016 outbreak according to date of symptom onset, origin of infection and area (outside or in slums). Fig C in S1 Results. Relative incidence (cases per 100,000 inhabitants) of autochthonous dengue cases according to the value of demographic covariates and area of the city (outside or in slums). Fig D in S1 Results. Monthly number of people entering Argentina according to bordering country from October 2015 to July 2016. Fig E in S1 Results. Posterior predictive check of hierarchical spatiotemporal modeling of dengue transmission. Fig F in S1 Results. Weekly global point-pattern spatiotemporal analysis of dengue cases according to different time rings ($\tau$) using the mark-connection function. Fig G in S1 Results. Duration of transmission foci according to the population within foci for different combinations of radii (columns) and time windows (rows). Fig H in S1 Results. Transmission foci detection under different radii and time windows (tau) in and outside slums. Fig I in S1 Results. Weekly detection of dengue transmission foci (in orange) according to local spatiotemporal analysis of autochthonous cases (black points) in Buenos Aires.
(PDF)

## Acknowledgments

The Servicio Meteorológico Nacional (Argentina) provided the daily weather data of its meteorological stations. Erin Mordecai kindly shared the posterior distributions of parameters to calculate $R_0$. Adrian Baddeley and the Spatstat team provided improvements in the package functions to run the analyses. Thorsten Wiegand gave helpful advice on the spatial analyses. We thank Dan Haydon for a critical review of the manuscript. The current research was undertaken as part of an agreement (No. 4352/17) between the Ministry of Health of the City of Buenos Aires, the University of Comahue, and CONICET.

## Author Contributions

**Conceptualization:** Juan Manuel Gurevitz, Juan Manuel Morales.

**Data curation:** Juan Manuel Gurevitz, Julián Gustavo Antman.

**Formal analysis:** Juan Manuel Gurevitz.

**Investigation:** Juan Manuel Gurevitz, Julián Gustavo Antman.

**Methodology:** Juan Manuel Gurevitz, Juan Manuel Morales.

**Resources:** Karina Laneri, Juan Manuel Morales.

**Software:** Juan Manuel Gurevitz.

**Supervision:** Juan Manuel Morales.

**Visualization:** Juan Manuel Gurevitz.

**Writing – original draft:** Juan Manuel Gurevitz.

**Writing – review & editing:** Juan Manuel Gurevitz, Julián Gustavo Antman, Karina Laneri, Juan Manuel Morales.

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
