## [Decision Letter · Decision Letter 0]

27 Jan 2021

Dear Dr. Gurevitz,

Thank you very much for submitting your manuscript "Temperature, traveling, slums, and housing drive dengue transmission in a non-endemic metropolis" for consideration at PLOS Neglected Tropical Diseases. As with all papers reviewed by the journal, your manuscript was reviewed by members of the editorial board and by several independent reviewers. In light of the reviews (below this email), we would like to invite the resubmission of a significantly-revised version that takes into account the reviewers' comments. 

We cannot make any decision about publication until we have seen the revised manuscript and your response to the reviewers' comments. Your revised manuscript is also likely to be sent to reviewers for further evaluation.

Sincerely,

Joseph T. Wu

Deputy Editor

Joseph Wu

Deputy Editor

Reviewer's Responses to Questions

**Key Review Criteria Required for Acceptance?**

**Methods**

-Are the objectives of the study clearly articulated with a clear testable hypothesis stated?

-Is the study design appropriate to address the stated objectives?

-Is the population clearly described and appropriate for the hypothesis being tested?

-Is the sample size sufficient to ensure adequate power to address the hypothesis being tested?

-Were correct statistical analysis used to support conclusions?

-Are there concerns about ethical or regulatory requirements being met?

Reviewer #1: This article describes the spatio-temporal pattern of DENV transmission in Buenos Aires, a large metropolis that sporadically experiences localized outbreaks. The overall design and proposed analyses are sound. However, I have some requests for clarification in the methods: 

1. Data: I feel authors should expand more on their geocoding strategy. Specifically, they mention that cases were geocoded to the house level (I assume any address not matched to the house was discarded, as it will show as overlapping points at a ‘high order’ unit, such as neighborhood or Zip code). 

This brings several questions, including:

a) Expand on how geocoding is done. 

b) how were cases geocoded inside slums? One attribute of slums is that they lack proper address systems, due to lack of registry. Including such data in any geocoding platform will lead to all points being matched to the 'neighborhood' or zipcode. Unless authors can explain this in more detail, I find some of the findings (higher aggregation in slums) to be potentially driven by a bias of the geocoding algorithm used. 

c) why only confirmed cases were included? In many countries, testing stops once an outbreak is declared (or a fraction of cases is confirmed, depending on resources). I suggest running analysis with all probable cases to confirm that patterns observed hold. Also, if cases were confirmed by lab, I assume there are cases that were discarded due to a lab test negative. Why did authors not use both as a case-control dataset, which may have allowed for more rigorous spatial analysis. 

d) Information about the time and source of census data is required. How many years elapsed since the census and the outbreak?

2. Analysis. In general, analyses are sound. Some more detail is needed to better understand their applicability. For instance:

e) Models appeared to run at the census tract level, but if one sees the distribution of cases it may be logical to expect that cases distribute following a zero-inflated negative binomial distribution. Please explain how zeros were handled. 

Also

Given that the main significant effects are not spatio-temporally varying, applying such an involved model (hierarchical spatio-temporal) may have given rise to similar results than a simpler spatial model with a random intercept. Perhaps I am wrong, but a justification of why spatio temporal models are applied in the presence of variables not varying ins space-time is required. Additionally, other approaches for adding spatial structure (more amenable for simulating human movement) such as correlated random walks may have shown different patterns emerge, compared to a simple 'queen' contiguity scheme. Please, justify your choice of spatial effects on your model. 

f) More explanation is needed to understand the parameter alpha in the model. From the description of methods and Suppl, it is not clear to me how alpha becomes a detection probability. The model is formulated at weekly spatiotemporal slices, and there is no explanation of how such observation aggregates to provide a total alpha for the city. I assume you are averaging detectability throughout the outbreak, but please clarify how the main effect in the model translates in a detection probability. How estimates change if all cases (rather than only those geocoded) are used? is alpha scaled linearly or exponentially to the increase of N? 

g) spatio-temporal analyses seem correct, but more detail is needed to justify the use of 'mark-connection functions' . Also, results from this function are not clearly shown (just in text, but no reference to actual values or test statistic). 

h) L246. THere is mention of random labeling of infected and not infected attributes of points, but all functions used (Local K) are based on cases only, unless the authors ran a bivariate local K function. Please explain.

Reviewer #2: At the beginning of the paper, the authors defined a clear objective: to describe the main dynamic/process of dengue transmission in a non-endemic location in Argentina using a spatial, social, and climatic approach. They designed a detailed methodology that is easily followed during reading, also provides a good description of the study zone and its corresponding variables. 

The methodology proposed seems to follow a clear line from the objective to the data analysis and presented results.

Even so, the paper presents only one gap in a method description, that I think could be fixed during the revision process (that I described in Editorial and Data Presentation Modifications)

Reviewer #3: (No Response)

**Results**

-Does the analysis presented match the analysis plan?

-Are the results clearly and completely presented?

-Are the figures (Tables, Images) of sufficient quality for clarity?

Reviewer #1: Analyses math what described in methods, and graphics are well designed. I just have some suggestions for improvement:

i) For R0, I believe there is a missed opportunity here. It would be good to find a way of determining statistically that there is an association between temperature and R0 being higher than 0. Cross correlation analyses would tell if both time series are associated, but other analyses may be more informative. 

j) As mentioned on previous section, explain the rationale of developing a hierarchical model, but more importantly describe findings in more detail. How well the model fitted the data? what is the distribution of random effects, are they spatially autocorrelated? Providing a measure of fit of the model to the data would help identify whether this is the most sound statistical approach to address the question of interest. 

k) spatio-temporal analyses (locak K) could be explained in more detail. Need to indicate how many cases were members of a cluster. What is the average (SD) size of clusters, duration, etc. And use such values to support claims of differences between slums and no slum areas. Also, indicate that foci are areas where locak K was significant (that was my interpretation of it, am I wrong?).

L) one of the figures in the supplement shows violin plots, and indicate mean values. Make sure you are not plotting medians, as that is what generally is shown in such plots. 

M) L352. this paragraph mentions results that cannot be found anywhere, and are too vague to find them informative. Where are those findings shown? You cannot report in results a value of ~300m. Have to provide actual value, and deviation around it if a measure of central tendency. 

M) a sensitivity analysis of the LocalK was performed, but the results are presented to vaguely. Need to report how many cases are added as clusters with each method, how clustering distances vary, etc, before claiming that both distances used are comparable..the supplementary figure is not enough. 

N) I disagree with the approach to quantify the 'cost of the intervention'. To me, this component is confusing. Given you selected which cases to geocode, the number of cases per area is a subset of what you could have. Then, unless you make sensitivity analyses around that metric (and call it something else), I do not see the value of estimating the proportion of cases in clusters to the proportion of the population measured by a census that likely occurred years before or after the outbreak. You may be under or over estimating this metrics in slums much more, as population estimates may be very variable over time. Plus, I have no idea how you arrived to the 300 m as the optimum. Again, this does not bring up any information that would strengthen the paper.

Reviewer #2: The authors have an extensive result section with informative tables and figures, which could improve the resolution in Fig.1 S2. For each of them presented in the main results, the paper has a complete readable explanation.

Reviewer #3: (No Response)

**Conclusions**

-Are the conclusions supported by the data presented?

-Are the limitations of analysis clearly described?

-Do the authors discuss how these data can be helpful to advance our understanding of the topic under study?

-Is public health relevance addressed?

Reviewer #1: O) In most part, I agree with the methods and findings. However, I have a concern that impacts some of the conclusions. The authors do not include any information about vector control actions undertaken to contain the outbreak. Without such information, and given Buenos Aires is not endemic for dengue, the authors cannot conclude that temperature drove the epidemic down. Likely, there is an impact of vector control (even if marginal) that may have led to an interruption of virus transmission in specific hotspot areas. Then, conclusions about temperature-driven outbreak extinction are not fully justified. The R0 model of mordecai assumes no vector control actions are conducted, which probably is not the case for this outbreak. Please, clarify. 

P) Need to add a paragraph outlining limitations of the study, which there are many. 

Q) the public health relevance can be emphasized more. How is this information able to help Buenos Aires respond to the next outbreak? Control in mega-cities is challenging, and I encourage the authros to use the discussion section to expand on possible outcomes.

Reviewer #2: The discussion presented in the paper is in concordance to the obtained results and reviewed literature and follows a clear structure. For instance, they determine that some socio-economic variables are correlated with the propagation of dengue over the study population, e.g., migration and some precarious conditions; by implementing the Hierarchical modeling and global spatial distributions.

Also, they suggested there were some limitations in their study like the implemented time series (only one dengue outbreak) that could cause that the rains do not present a direct correlation with dengue cases.

Finally, they suggested that this research methodology would be implemented for other infectious diseases transmitted by the Aedes aegypti, e.g., Zika and Chikungunya; for the design of public health strategies.

Reviewer #3: (No Response)

**Editorial and Data Presentation Modifications?**

Reviewer #1: Minor revision. 

R) I encourage authors to cite and review the two references below, as they outline the importance of spatial heterogeneity for dengue stratification efforts, including the overlap between dengue and other Aedes-borne viruses: https://iris.paho.org/handle/10665.2/51652 ; https://journals.plos.org/plosntds/article?id=10.1371/journal.pntd.0006298 . 

S) avoid overutilizing the word 'putative'.

Reviewer #2: 1. Instead of the robust/rigorous statistical method in lines 34 and 108, you might provide a brief mention of the statistical method implemented to make it clearer to the reader, e.g., a robust spatiotemporal regression.

2. You mentioned in lines 118-12 that some characteristics distinguish the ‘slums’; in fact, in lines 190-193 you said that you excluded these areas to implement the hierarchical spatiotemporal model in Buenos Aires. I would like to know how you could determine which zone is a slum. There is an index or a specific factor to delimiter these zones? Please make the way you classify a slum and a standard neighborhood clearer in the methodology section.

3. I suggest to take care of the vocabulary used for some assertions related to the economic state of the neighborhoods (the presented in lines 118,190-193, 500, and so on). For example, the lines 190-193 might be re-written as "Slums were excluded because of their distinctive characteristics associated with overall poor health systems. Those exclusions allowed for a better understanding of the transmission factors of DENV in the city"

Reviewer #3: (No Response)

**Summary and General Comments**

Reviewer #1: Overall, a good article. Some suggestions for improvement, which primarily focus on simplifying methods and addressing limitations of analyses and data:

T) Keep in mind that the factors you identified in your model are a proxy for the process of virus transmission, which is dependent on mosquitoes density, susceptibility of the human population, and suitability for dengue introduction. One may find income is a proxy in some studies, or piped water in others. Population density is quite consistent as an important predictor, in your case because you are using case counts and more populous census tracts will have a higher probability of having a case. I consider many of the associations with census data to be dependent on the temporal scale of census information (how 'recent' it is) and the spatial unit of aggregation (how fine scale you are analyzing). I recommend that authors state the limitations and make sure to follow up in future research by validating their statistical findings with field information. THis brings me to the next point:

U) A key missing piece of data is entomology. Authors should make an effort to discuss this gap, as there is a network of ovitraps that monitor mosquito eggs on a fine spatial and temporal frequency. Why are those data not included? Validating that hotspots are areas with high mosquito density would have strengthened the article.

V) more generally, I would like authors to revisit some of hte metrics provided (alpha and cost of intervention), add more detail on the methods used and the report of results, and discuss in more detail the limitations of their study, as well as the public health relevance. 

W) make your data available.

Reviewer #2: (No Response)

Reviewer #3: PNTD-D20–01700

General comments:

In this paper, the authors model the spatiotemporal dynamics of a dengue epidemic in Buenos Aires, investigating the effect of weather and sociodemographic factors on dengue incidence. The paper seems to have 3 separate sub-sections: a temporal analysis of the relationship between R0(T) and relative dengue incidence, a spatial analysis investigating the effect of sociodemographic factors, and an analysis of the cost of interventions. My main comment is that these three analyses aren’t currently tied together very coherently and the presentation of the methods lacks clarity, to the extent that the objective and relationship between these three different components is unclear.

Major comments:

1. Line 138 – could the authors provide definitions of confirmed, probably, suspected and discarded cases?

2. Line 144 – only confirmed cases were used in the analysis. Is there spatial variation in testing or healthcare usage that might affect the distribution of confirmed vs suspected cases?

3. Line 146 – please specify which weather variables were used. The data were obtained from the airport’s weather station. Can the authors also comment on how representative this one station is of the wider Buenos Aires metropolitan area? In particular, rainfall may be quite localised, even within a relatively small urban area.

4. The methods section could be presented more clearly. Initially, I was under the impression that the authors are using R0(T) to model variations in dengue incidence across space and time, but it seems this is not the case. Figure 3 seems to be the only place where R0(T) data are presented, as a global comparison of the time series. Even here, it’s not clear to me that there’s a clear correspondence between the weekly IRR and R0(T). I imagine this is because the authors only account for the weekly relative incidence, even though the effect of R0 may be lagged by several weeks. 

5. In their spatial model, the authors assume a Poisson distribution for the dengue case count, but as far as I can see do not provide any evidence to show that this is a reasonable assumption. Dengue incidence data are often overdispersed so it would be useful to provide a stronger justification for this model choice.

6. In this model, the authors consider cases in the previous week in the same census fraction, cases in the previous week in neighbouring fractions, and cases in the previous week city-wide. But it’s not clear to me whether the authors are aiming to model a biological process or statistical autocorrelation here. I would not expect cases in the previous week to be directly relevant to case incidence in the current week, because this is shorter than the extrinsic and intrinsic incubation periods. A biologically plausible model would need to account for the cumulated effect of cases, or preferably transmissibility, some weeks previously. So it seems to me that the authors are just using their model to account for correlation in their spatially stratified time series. If this is the case, then I find it much less compelling, and I would also want to see some evidence that a one week lag is sufficient to account for temporal and spatial correlation in the time series, before being convinced that the effects of other sociodemographic variables are free from confounding.

7. A major assumption seems to be that cases were infected at their census tract of residence. Is there evidence to show that most infections in Buenos Aires occur in domestic settings or close to the place of residence?

8. Line 253 – the authors estimate some measure of the cost of interventions, but this section is not very clear at all. What exactly is being assessed here and where do the cost data come from? The relevance of this part to the rest of the manuscript is not immediately obvious.

9. Minor comments:

10. Line 80 – the authors mention the inconsistent results regarding factors associated with social and spatial clustering of dengue between different studies. But I think some of these inconsistencies are explained by the level of transmission in different settings. In non-endemic settings, more densely populated areas with less effective vector control are more likely to be affected first. In places that subsequently support endemic transmission, it also means that populations in these areas may gain higher levels of herd immunity over time, such that an inverse association between, say, socioeconomic level and dengue risk is seen, as has been observed in some urban settings in Brazil.

PLOS authors have the option to publish the peer review history of their article (what does this mean?). If published, this will include your full peer review and any attached files.

Reviewer #1: No

Reviewer #2: No

Reviewer #3: Yes: Clarence Tam
---

## [Decision Letter · Decision Letter 1]

11 May 2021

Dear Dr. Gurevitz,

We are pleased to inform you that your manuscript 'Temperature, traveling, slums, and housing drive dengue transmission in a non-endemic metropolis' has been provisionally accepted for publication in PLOS Neglected Tropical Diseases.

Best regards,

Kendall McKenzie

Staff Admin

Joseph Wu

Deputy Editor

Reviewer's Responses to Questions

**Key Review Criteria Required for Acceptance?**

**Methods**

-Are the objectives of the study clearly articulated with a clear testable hypothesis stated?

-Is the study design appropriate to address the stated objectives?

-Is the population clearly described and appropriate for the hypothesis being tested?

-Is the sample size sufficient to ensure adequate power to address the hypothesis being tested?

-Were correct statistical analysis used to support conclusions?

-Are there concerns about ethical or regulatory requirements being met?

Reviewer #1: Authors addressed all my comments.

Reviewer #2: (No Response)

**Results**

-Does the analysis presented match the analysis plan?

-Are the results clearly and completely presented?

-Are the figures (Tables, Images) of sufficient quality for clarity?

Reviewer #1: Authors addressed all my comments.

Reviewer #2: (No Response)

**Conclusions**

-Are the conclusions supported by the data presented?

-Are the limitations of analysis clearly described?

-Do the authors discuss how these data can be helpful to advance our understanding of the topic under study?

-Is public health relevance addressed?

Reviewer #1: Authors addressed all my comments.

Reviewer #2: (No Response)

**Editorial and Data Presentation Modifications?**

Reviewer #1: Authors addressed all my comments.

Reviewer #2: Accept

**Summary and General Comments**

Reviewer #1: Authors addressed all my comments.

Reviewer #2: The authors have addressed all the concerns and suggestions raised by reviewers. I recommend its acceptance.

PLOS authors have the option to publish the peer review history of their article (what does this mean?). If published, this will include your full peer review and any attached files.

Reviewer #1: No

Reviewer #2: No

---

## [Editor Report · Acceptance letter]

8 Jun 2021

Dear Dr. Gurevitz,

We are delighted to inform you that your manuscript, "Temperature, traveling, slums, and housing drive dengue transmission in a non-endemic metropolis," has been formally accepted for publication in PLOS Neglected Tropical Diseases.

Best regards,

Shaden Kamhawi

co-Editor-in-Chief

Paul Brindley

co-Editor-in-Chief
